# Global Proteome Profiling of the Temporal Cortex of Female Rats Exposed to Chronic Stress and the Western Diet

**DOI:** 10.3390/nu14091934

**Published:** 2022-05-05

**Authors:** Maria Marta Nowacka-Chmielewska, Daniela Liśkiewicz, Arkadiusz Liśkiewicz, Marta Przybyła, Łukasz Marczak, Anna Wojakowska, Konstancja Grabowska, Mateusz Grabowski, Jarosław Jerzy Barski, Andrzej Małecki

**Affiliations:** 1Laboratory of Molecular Biology, Institute of Physiotherapy and Health Sciences, Academy of Physical Education, 40-065 Katowice, Poland; d.liskiewicz@awf.katowice.pl (D.L.); a.malecki@awf.katowice.pl (A.M.); 2Department of Physiology, Faculty of Medical Sciences in Katowice, Medical University of Silesia, 40-055 Katowice, Poland; adliskiewicz@gmail.com (A.L.); jbarski@sum.edu.pl (J.J.B.); 3Department for Experimental Medicine, Faculty of Medical Sciences in Katowice, Medical University of Silesia, 40-055 Katowice, Poland; mgrabowska@sum.edu.pl (M.P.); grabowska.i.konstancja@gmail.com (K.G.); mateusz.m.grabowski@gmail.com (M.G.); 4Institute of Bioorganic Chemistry, Polish Academy of Sciences, 61-704 Poznań, Poland; lukasmar@ibch.poznan.pl (Ł.M.); astasz@man.poznan.pl (A.W.)

**Keywords:** Western diet, chronic stress, brain proteome, female rats

## Abstract

The increasing consumption of highly processed foods with high amounts of saturated fatty acids and simple carbohydrates is a major contributor to the burden of overweight and obesity. Additionally, an unhealthy diet in combination with chronic stress exposure is known to be associated with the increased prevalence of central nervous system diseases. In the present study, the global brain proteome approach was applied to explore protein alterations after exposure to the Western diet and/or stress. Female adult rats were fed with the Western diet with human snacks and/or subjected to chronic stress induced by social instability for 12 weeks. The consumption of the Western diet resulted in an obese phenotype and induced changes in the serum metabolic parameters. Consuming the Western diet resulted in changes in only 5.4% of the proteins, whereas 48% of all detected proteins were affected by chronic stress, of which 86.3% were down-regulated due to this exposure to chronic stress. However, feeding with a particular diet modified stress-induced changes in the brain proteome. The down-regulation of proteins involved in axonogenesis and mediating the synaptic clustering of AMPA glutamate receptors (Nptx1), as well as proteins related to metabolic processes (Atp5i, Mrps36, Ndufb4), were identified, while increased expression was detected for proteins involved in the development and differentiation of the CNS (Basp1, Cend1), response to stress, learning and memory (Prrt2), and modulation of synaptic transmission (Ncam1, Prrt2). In summary, global proteome analysis provides information about the impact of the combination of the Western diet and stress exposure on cerebrocortical protein alterations and yields insight into the underlying mechanisms and pathways involved in functional and morphological brain alterations as well as behavioral disturbances described in the literature.

## 1. Introduction

Worldwide, the excessive consumption of highly processed foods with a high amount of saturated fatty acids, simple carbohydrates, and calorie density has increased considerably [1,2]. A variety of palatable high-fat and/or high-carbohydrate diets (HFDs or cafeteria or Western-pattern diets) have been applied in experimental research to study the consequences of unhealthy eating and their association with overweight, obesity, metabolic complications [2,3], and neurodegenerative diseases as well as behavioral changes in humans and laboratory rodents [4,5]. Animal studies have confirmed that the long-term intake of obesogenic diets impairs cognition, especially with regard to aspects that are dependent on the hippocampus, including memory processes and reversal learning [6,7]. There is also emerging evidence from clinical studies that high energy diets—in addition to an increased risk of cardiovascular episodes, type 2 diabetes, and dyslipidemia—may lead to the development of neurodegenerative and psychiatric diseases [8,9].

Chronic stress, even daily life-related stress of moderate-intensity, is widely acknowledged as a predisposing or precipitating factor in neuropsychiatric diseases, especially in women [10]. There is a clear relationship between disturbances induced by, especially long-lasting, stressful stimuli and cognitive deficits in rodent models of affective disorders [11]. For laboratory animals living in groups, the disturbance of their social environment is a more significant stress-inducing factor than physical stimuli [12,13]. Accumulating evidence indicates that metabolic and mood disturbances are strongly interrelated, reviewed in [14,15,16]. The behavioral effects of chronic stress are reportedly ameliorated [17] or exacerbated [18] with high-fat and/or high-carbohydrate diets. In addition, a previous study suggested that a combination of HFD and chronic stress may act synergistically, aggravating insulin resistance associated with the abnormal hypothalamic–pituitary–adrenocortical (HPA) axis [19]. Despite the growing evidence supporting the important role of overweight and obesity in the development of mood disorders, it is not fully understood how high-fat and/or high-carbohydrate diets and metabolic disturbances interact in response to chronic stress.

The entanglement of obesity and stress described above demonstrates a critical need to assess how obesogenic diets and stress can modulate central metabolic and behavioral outcomes in obesity. Finally, pervasive sex differences in metabolic traits, such as body fat distribution, glucose homeostasis, insulin signaling, ectopic fat accumulation, and lipid metabolism, have often been omitted in human and animal model research [20,21]. Considering that susceptibility to mental and metabolic diseases is strongly associated with sex [22,23] and exposure to environmental and lifestyle factors [24], we decided to perform our study on female rats.

## 2. Materials and Methods

### 2.1. Animals and Experimental Design

Female Long–Evans rats with or without exposure to social stress received a standard chow diet (Lobofeed B, Morawski) or an obesogenic rodent diet (the Western diet with human snacks, described below) and water ad libitum. Rats, 9 weeks old at the start of the 12-week intervention, were housed 3–4 per cage in a climate-controlled room (22 ± 2 °C, relative humidity: 55 ± 10%) with a 12:12-h light/dark cycle starting at 07:00 A.M. All animals were provided by the Animal House of the Department for Experimental Medicine, Medical University of Silesia (Katowice, Poland), and were treated in accordance with Directive 2010/63/EU for animal experiments using the protocols approved and monitored by the Local Ethics Committee for Animal Experimentation in Katowice (approved protocol number). The minimum number of rats required to obtain consistent data was used, and every effort was taken to minimize the suffering of the animals.

The rats were randomly divided into 4 experimental groups. Upon the initiation of the experiments, the rats were fed ad libitum with standard chow (control group, CTR, *n* = 12) or the same chow supplemented with Western diet snacks (WD group, *n* = 12; refer to the Obesogenic Rodent Diet section for more details) for 12 weeks in order to induce weight gain. The stress group consisted of rats fed with the standard chow and subjected to a procedure of chronic stress (CS group, *n* = 6) induced by chronic social instability (detailed below). Animals in the WD/CS group (*n* = 6) were fed with the Western diet and subjected to chronic stress. In each group, all foods were provided ad libitum. All animals had free access to water for the duration of the experiment. Body weight (g) and energy intake (kcal/day) were measured daily. The rats were decapitated on the day following the completion of the diet and stress procedures. Cortical samples and blood were collected for further measurements.

### 2.2. Obesogenic Rodent Diet

To mimic the human obesogenic diet, animals were fed with commercially available human snacks as previously described [25]. The rats received one of two sets of snacks interchangeably. These were given to the animals on alternate days as one diversified diet. Set 1 included the following: candy bar (Mars; Mars Inc., McLean, VA, USA), crackers (Lajkonik Snacks, Skawina, Poland), and kabanos (dry sausage made of pork; Tarczyński, Trzebnica, Poland). The following snacks comprised set 2: candy bar (Bounty; Mars Inc., McLean, VA, USA), potato chips (Lays Salt; PepsiCo, Raleigh, NC, USA), and Tilsit cheese (Hochland SE, Heimenkirch, Germany). WD rats received clean water and a sweet beverage, a 10% fructose solution (39.8 kcal/100 mL; sweetened with Consweet Sweetener Confex-Product, Warsaw, Poland), in a second container. The average caloric density of these 2 dietary sets (including 10% fructose) was 4.84 kcal/g with the following caloric profile: carbohydrates 33.2%, fat 33.1%, and protein 16.6%. The standard chow diet energy content (3.57 kcal/g) comes from 67% carbohydrates, 25% protein, and 8% fat. In each group, all foods were provided ad libitum. All animals had free access to water for the duration of the experiment. The food was supplied daily, and food intake was monitored each day (the chow and snacks were weighed before and after consumption). Liquid intake (water and 10% fructose) was monitored every second day. Total energy intake was determined by calculating the combined intake of liquid fructose, snacks, and chow. The composition and nutritional profile of the diets are provided in the Appendix A.

### 2.3. Chronic Social Stress Paradigm

The chronic social stress paradigm was adapted from the protocol described previously [25,26]. In the present study, the authors extended the duration of the stress procedure to 12 weeks. As an unstable social situation and isolation are strong stressors for female rats, uncontrollability is modeled in the chronic stress paradigm by alternating the isolation and crowding phases. From the beginning of the experiment, the animals subjected to stress were kept alone in a cage (36 cm × 20 cm × 15 cm), excluding 3 or 6 h (08:00 A.M.–11:00 A.M., or 08:00 A.M.–02:00 P.M.), during which the rats stayed in groups of 6 in cages with a smaller area (30 cm× 20 cm × 15 cm). To enhance unpredictability, marked rats were mingled with unfamiliar animals in each crowding phase. Rats from the CS and WD/CS groups were subjected to the same stress procedure. The animals in the other groups stayed in groups of 5–6 in standard cages (52 cm × 31 cm × 19 cm).

### 2.4. Glucose Tolerance Test (GTT)

Rats underwent an intraperitoneal (i.p.) glucose tolerance test (GTT) after 12 weeks of experiments. Blood collection began at 8:00 A.M. Rats were fasted overnight, and a baseline blood glucose concentration was measured in tail blood (CardioCheck Plus Professional, PTS Diagnostics, Whitestown, IN, USA). Then, the rats were injected i.p. with 20% glucose (2 g/kg body weight), and blood glucose concentrations were measured at 15, 30, 60, and 120 min post-injection.

### 2.5. Tissue Collection

Rats were fasted overnight before tissue collection. The body weight of each animal was measured. The collection of blood and brain samples began at 7:00 A.M. Due to hormonal fluctuations, rats were sacrificed alternately (one animal from each group) during the section. The rats were decapitated, and trunk blood was collected. Afterward, the brain was rapidly removed and placed on an ice-chilled metal plate, ventral side up. The brain was turned dorsal side up and hemisected. Subsequently, an approximately 2.0 mm-thick slice of the temporal cortex was cut off from each hemisphere. The cerebrocortical samples from each rat were immediately frozen on dry ice and stored at −80 °C for liquid chromatography-tandem mass spectrometry (LC-MS/MS) analysis. The blood samples were centrifuged at 2500× *g* for 10 min (4 °C). Then, the collected serum was stored at −80 °C for subsequent analysis. The average adipose tissue content in the animals was assessed by weighing the retroperitoneal and subcutaneous (from the abdominal region) adipose tissue. Then, the liver, heart, and kidney were weighed. After the removal of the internal organs, fat, skin, and tail, the trunk of the decapitated animal was weighed and considered as lean body mass.

### 2.6. Biochemical and Hormonal Assays

Biochemical and hormonal assessments were performed on the serum samples from the following groups: CTR (*n* = 8), WD (*n* = 8), CS (*n* = 6), and WD/CS (*n* = 6). The serum glucose and lipid profiles were measured using a Mindray BS-200 Chemistry Analyzer (Shenzhen Mindray Bio-Medical Electronics Co., Shenzhen, China). The level of serum 17beta-estradiol was determined with a chemiluminescent immunometric assay on an IMMULITE 2000 analyzer (Siemens Healthcare Diagnostics, Eschborn, Germany). The serum concentrations of insulin and leptin were measured using commercial ELISA kits according to the manufacturer’s instructions (Leptin: Thermo Fisher Scientific, Waltham, MA, USA, intra-assay coefficient of variation (%CV): 4.3 %, and the sensitivity of the assay was 22 pg/mL; Insulin: R & D Systems, Minneapolis, MN, USA, % CV: <10%, the sensitivity was 5 μlU/mL). The absorbance at 450 nm was measured using a microplate reader (Biotek Synergy LX, Agilent, Santa Clara, CA, USA).

Insulin sensitivity was calculated in the serum samples from the rats by means of the quantitative insulin sensitivity index (QUICKY = 1/(log(I) + log(G)), where I is the fasting serum insulin concentration, and G is the fasting glucose concentration [27].

### 2.7. Proteomic Analysis

LC-MS/MS was performed on rat temporal cortex samples from groups; (i) CTR (*n* = 11), (ii) WD (*n* = 11), (iii) CS (*n* = 5), and (iv) CS/WD (*n* = 6). The frozen brain tissue samples were ground in liquid nitrogen in precooled adaptors for 45 s at a 30 Hz frequency using a ball mill MM400 (Retsch, Haan, Germany). Then tissues were lysed in a buffer with 1 M triethylammonium bicarbonate (TEAB) and 0.1% sodium dodecyl sulfate (SDS) and automatically homogenized using a Precellys 24 homogenizer (Bertin Technologies, Montigny-le-Bretonneux, France) in 0.5-mL tubes pre-filled with ceramic (zirconium oxide) beads (Bertin Technologies, Montigny-le-Bretonneux, France). Next, the material was subjected to a threefold cycle of freezing and thawing. Then, the tissue in the buffer was sonicated in a bath for 3 1-min cycles on ice and homogenized again using the Precellys 24 instrument. The protein concentration was measured using a Pierce BCA protein assay kit (Thermo Fisher Scientific, Waltham, MA, USA) in the isolated protein fraction according to the manufacturer’s instructions.

Next, 10-microgram aliquots of proteins were diluted with 15 µL of 50 mM NH_4_HCO_3_ and reduced with 5.6 mM DTT for 5 min at 95 °C. Samples were then alkylated with 5 mM iodoacetamide for 20 min in the dark at RT. The proteins were digested with 0.2 µg of sequencing-grade trypsin (Promega, Madison, WI, USA) overnight at 37 °C.

The analysis was performed with the use of the Dionex UltiMate 3000 RSLC nanoLC System connected to a Q Exactive Orbitrap mass spectrometer (Thermo Fisher Scientific, Waltham, MA, USA). Peptides derived from the in-solution digestion were separated on a reverse-phase Acclaim PepMap RSLC nanoViper C18 column (75 µm × 25 cm, 2 µm granulation) using an acetonitrile gradient (from 4 to 60%, in 0.1% formic acid) at 30 °C and a flow rate of 300 nL/min (for 230 min). The spectrometer was operated in data-dependent MS/MS mode with survey scans acquired at a resolution of 70,000 at *m*/*z* 200 in MS mode and 17,500 at *m*/*z* 200 in MS^2^ mode. The spectra were recorded in the scanning range of 300–2000 *m*/*z* in the positive ion mode. Higher energy collisional dissociation (HCD) ion fragmentation was performed with normalized collision energies set to 27.

### 2.8. Data Analysis of Proteins

Protein identification was performed using the Swiss-Prot rat database with a precision tolerance set to 10 ppm for peptide masses and 0.08 Da for fragment ion masses. All raw data obtained for each dataset was imported into MaxQuant ver. 1.5.3.30 for protein identification and quantification. A protein was considered as positively identified if at least 2 peptides per protein were found by the Andromeda search engine and the peptide score reached the significance threshold FDR = 0.01.

The obtained data were exported to Perseus ver. 1.5.3.2 software (part of the MaxQuant package, Max-Planck Institute of Biochemistry, Computational Systems Biochemistry, Munich, Germany). The numeric data were transformed into a logarithmic scale, and each sample was annotated with its group affiliation. Next, the data were filtered based on valid values–proteins. Samples that had valid values in 70% of at least one group were kept in the table. A one-way analysis of variance (ANOVA) analysis was performed on the analyzed sample data with permutation-based FDR 0.05 used for truncation, and the resulting list of differentiating proteins was normalized using a Z-score algorithm for the hierarchical clustering of data.

### 2.9. Gene Ontology (GO) Term Enrichment Analysis

The gene names of significantly regulated proteins (q < 0.05 and fold-change of +/−50%) were submitted to a PANTHER Overrepresentation Test online (http://www.pantherdb.org, accessed on 2 February 2022) using PANTHER version 16.0 (released 16 November 2021). The PANTHER functional annotation tool [28,29] and Metascape [30] were used to identify the significant ontology associations. The complete list of *Rattus norvegicus* proteins detected by mass spectrometry-based proteomics was used for the background, while the GO term subcategories “GOTERM_BP_DIRECT”, “GOTERM_CC_DIRECT”, and “GOTERM_MF_DIRECT” were selected for analysis. The false discovery rate (FDR) correction of Fisher’s *p*-value was applied to identify the significant functional annotations (significance was considered when *p* < 0.05).

### 2.10. Statistical Analysis of Non-Proteome Data

Statistical analysis was performed using the GraphPad Prism 9.3.1 software (GraphPad Software Inc., San Diego, CA, USA). The distribution of each data set was checked for normality using the Shapiro–Wilk test. To analyze repeated measures of data, two-way ANOVA with Tukey or Sidak’s multiple comparison test or the REML mixed-effects model (in case of missing values) was applied. Apart from proteomics data, other data were expressed as the mean ± standard error (SD). In all analyses, *p*-values of less than 0.05 were considered to be statistically significant. Box plots with whiskers (median and min.–max.) were exceptionally used to present the proteomic data.

## 3. Results

### 3.1. Phenotypic and Biochemical Parameters

CTR and CS rats fed with the Western diet consumed an average of 40% more calories (two-way ANOVA: diet F (1, 104) = 56.46, *p* < 0.0001; post hoc test: CTR vs. WD: *p* < 0.0001, CTR vs. WD/CS: *p* < 0.0001; CS vs. WD/CS: *p* < 0.0001, Table 1) and gained weight more excessively than those fed with a standard diet (mixed-effects model, intervention F (3, 32) = 23.01, *p* < 0.0001, time F (2.686, 78.79) = 294.2, *p* < 0.0001, intervention x time F (36, 352) = 15.41, *p* < 0.0001, Appendix A). WD and WD/CS rats were ~31% and ~17% heavier, respectively, compared to standard chow-fed rats (CTR vs. WD: *p* < 0.0001; CS vs. WD/CS: *p* < 0.05) (Table 1). These observations correspond with changes in adipose tissue masses (two-way ANOVA: diet F (1, 32) = 66.13, *p* < 0.0001, stress F (1, 32) = 15.63, *p* = 0.0004, diet x stress F (1, 32) = 8.13, *p* = 0.0075). Being fed with the Western diet resulted in a 3-fold increase of adipose tissue content expressed as a % of total body weight (CTR vs. WD: *p* < 0.0001), and a 2-fold increase in the stressed rats (CS vs. WD/CS: *p* < 0.0001), suggesting that the 12-week consumption of the Western diet induced an obese phenotype in female rats and also showing that exposure to stress modifies Western diet-induced phenotypic changes (Table 1).

The consumption of the Western diet resulted in a significant elevation of glucose levels only in the stressed rats (two-way ANOVA: diet (F (1, 20) = 10.32; *p* = 0.0048), stress (F (1,20) = 14.14, *p* = 0.0012), diet x stress interaction (F (1, 20) = 6.29, *p* = 0.02); post hoc test: CTR vs. WD/CS: *p* < 0.001; please see Table 2 for details). For the GTT, statistical analysis revealed a significant interaction between intervention and time (F (12, 128) = 2.10, *p* = 0.02), with effects for time (F (2.762, 88.39) = 295.2, *p* < 0.0001) but not for intervention alone (F (3, 32) = 2.40, *p* = 0.08). At the final point of GTT, the highest glucose concentration was observed in the WD/CS group compared to control rats (*p* = 0.0069) (Appendix A). Being fed with the Western diet increased serum leptin levels by 7 times (two-way ANOVA: diet F (1, 24) = 27.51; *p* < 0.0001, stress F (1, 24) = 32.04; *p* < 0.0001, diet x stress F (1, 24) = 77.05; *p* < 0.0001; post hoc test: CTR vs. WD: *p* < 0.0001), but the level of this hormone in the WD/CS group did not significantly differ from that in the CTR group. The examination of the serum insulin levels showed no significant effects (two-way ANOVA: diet: (F (1, 23) = 0.81, *p* = 0.37), stress (F (1, 23) = 0.042, *p* = 0.83). However, a marker of insulin sensitivity, QUICKI, was calculated, thus demonstrating a significant effect of the diet (F (1, 19) = 4.38; and stress-impaired insulin sensitivity (CTR vs. WD/CS: *p* = 0.013).

For the lipid profile, a significant effect of diet exposure (F (1, 19) = 24.31, *p* < 0.0001) was noted only in the LDL cholesterol levels. The total cholesterol levels tended to increase in the WD group; however, these effects were not statistically significant. Finally, to assess the potential influence of estradiol levels on the studied factors, we measured the serum level of this hormone at the end of the experiment. We did not observe statistically significant differences between the groups (Kruskal–Wallis test, *p* = 0.27, Table 2).

### 3.2. LC-MS/MS Analysis of Differentially Expressed Proteins

Based on the discovery analysis, a total of 2793 proteins were identified and quantified, of which 239 were significantly different between the temporal cortices across the 4 animal groups. An overview of the changes induced by the diet, stress, and a combination of stress and diet across all animals was presented as a heat map of the normalized label-free quantitation intensity (LFQ) for all detected proteins, which significantly differed between groups (Appendix A). To determine which proteins were significantly altered by either the diet or the stress exposure, a two-way ANOVA with Dunnett’s multiple comparison test was conducted for each protein detected in this proteomic dataset. Statistical analysis showed that consuming the Western diet resulted in changes only in 5.4% of proteins (13/239) as compared to the control rats. Of those 13 proteins, 10 were up-regulated and 3 down-regulated. Almost half of the identified proteins (117/239) were affected by chronic stress exposure, and this exposure down-regulated 86.3% of them (101/117). The Western diet modulated the stress-induced changes; namely, 61% of identified proteins (146/239) were changed in the WD/CS group in comparison to CS rats, and the vast majority of those proteins showed an increased expression (126/146) (Appendix A shows the fold change of the identified proteins with the adjusted *p*-value). Among the most significantly down-regulated (fold change < 0.45) proteins in response to stress were those involved in autophagy (Map1lc3a, Microtubule-associated proteins 1A/1B light chain 3A), axonogenesis (Nptx1, Neuronal pentraxin-1), or cell differentiation (Hbb-b1, Hemoglobin subunit beta-1; Wdr7, WD repeat-containing protein 7, TGF-beta resistance-associated protein TRAG), as well as proteins related to metabolic processes: Mrps36 (28S ribosomal protein S36, mitochondrial), Ndufb4 (NADH dehydrogenase [ubiquinone] 1 beta subcomplex subunit 4), and Rps12 (40S ribosomal protein S12) (please see box plots for representative proteins in Figure 1A). Panel B shows the representative stress-induced proteins based on the fold change (>1.9). This included Basp1 (Brain acid soluble protein 1), Cend1 (Cell cycle exit and neuronal differentiation protein 1) involved in the central nervous system (CNS) development and differentiation, Prrt2 (Proline-rich transmembrane protein 2) engaged in synaptic transmission in the CNS and the final steps of neurotransmitter release in hippocampal neurons, and Ncam1 (Neural cell adhesion molecule 1) involved in the multicellular organismal response to stress and the modulation of synaptic transmission. Appendix A shows the raw MS data for all identified proteins and summarizes the information for these proteins, including accession number (IDs), gene, full protein name, and molecular weight.

### 3.3. Gene Ontology (GO) Analysis

By means of GO analysis, the biological processes (GO:BP), cellular component (GO:CC), and molecular function (GO:MF) of proteins were annotated. As Figure 2A shows, the most overrepresented biological processes were related to the generation of precursor metabolites and energy, synapse transport, and the regulation of neurotransmitter levels. In the CC annotation, the highly enriched terms included the distal axon, mitochondrial membrane, postsynapse, and glutamatergic synapse (Figure 2B). The molecular function annotation was consistent with GO:BP, including nucleoside-triphosphatase and proton-transporting ATP synthase activities (Figure 2C).

Organizing the proteomic data by interventions also allowed the determination of which biological processes changed by chronic stress or the Western diet alone were also affected by the combination of these two interventions. There was an overlap between proteins regulated by the Western diet and chronic stress in GO terms, as the 20 biological pathways enriched in stress-regulated proteomes were enriched to a similar extent with WD rats, including neuron projection morphogenesis, the generation of precursor metabolites and energy, the regulation of postsynaptic membrane neurotransmitter receptor levels, proton transmembrane transport, and vesicle-mediated transport in the synapse (Figure 3A,B). Appendix A shows the data for all enrichment and annotation analyses.

### 3.4. KEGG and Protein-Protein Interaction Analyses

To reveal the pathways involved in proteins that were differentially expressed between the four experimental groups, KEGG pathway analysis was employed. As shown in Figure 4, the three most significantly enriched pathways were Huntington’s, Parkinson’s, and Prion diseases. Calmodulin 2 (Calm2), tubulins (Tubb4a, Tuba1a), Alpha-synuclein (Snca), and proteins involved in response to oxidative stress and mitochondrial metabolism (cytochrome c oxidase, ATP synthase, and NADH dehydrogenase subunits) were enriched in those pathways. The pathways of neurodegeneration/multiple diseases and diabetic cardiomyopathy were also significant pathways and shared altered proteins such as Atp2a32, Uqcr10, Ndufa12, Vdac1, and Vdac3. In order to further comprehend the interactions among the altered proteins, pathway and process enrichment analyses have been carried out with the following KEGG Pathway.

As shown in Figure 5, the most protein–protein interactions were observed between proteins associated with carbon metabolism, Parkinson’s disease, and diabetic cardiomyopathy pathways.

## 4. Discussion

In the present study, we used cerebrocortical proteome analysis to characterize the interaction between long-term exposure to social stress and the consumption of the Western diet in female rats. Our goal was to identify the proteins that were changed by chronic stress and to assess how consuming the Western diet modulates these changes. In total, 2793 proteins were identified, of which 239 proteins were found to be differentially expressed. Some of these proteins have been associated with multiple pathways of neurodegeneration, oxidative phosphorylation, and metabolic processes (glycolysis and gluconeogenesis).

Human studies have demonstrated that some people reporting greater stress exposure lose their appetite while others show an increased preference for high-fat and high-sugar foods [31,32]. In rodents, most studies have shown that stress reduces food intake, which is in line with human studies demonstrating ‘comfort’ eating in response to stressful situations [33,34]. Here, stress-modified diet-induced obese phenotypic parameters (animal weight and adipose tissue content) and some endocrine features related to obesity. Furthermore, exposure to 12 weeks of chronic social stress blunted the Western diet-induced increase in circulating levels of leptin and the body’s adipose mass. The reported effects of dietary interventions in combination with stress on serum leptin levels remain mixed, showing increased leptin levels [35,36] or reduced leptin elevations with high-fat feeding [37]. Previously, Chuang et al. (2010) showed that chronic social defeat stress stimulates the activity of the sympathetic nervous system via beta(3)-adrenergic receptors, and thereby causes lipolysis leading to the reduction in fat stores and leptin secretion [38]. It was also suggested that a decrease in leptin levels is involved in an adaptive response to stress exposure that, to some extent, can protect from the detrimental behavioral effects caused by chronic stress [38]. Surprisingly, consuming the Western diet in combination with the exposure to stress had a synergistic effect on serum glucose levels as well as on insulin resistance but did not change the serum glucose or insulin levels in non-stressed rats. A significant interaction between diet and stress was also observed in the GTT. At the final time point (120 min) of the GTT, the highest glucose concentration was observed in the WD/CS rats compared to the control (5.22 vs. 7.05 mmol/L). The additive or synergistic impairment of metabolism with WD/CS is in line with reports showing that stress worsens glucose homeostasis in animals fed a hypocaloric diet [35,39].

This study provides a few findings regarding changes in the brain proteome induced by the combination of the Western diet and stress. First, the cerebrocortical expression of proteins in the stressed rats was clearly altered, while negligible changes were detected in the brain proteome as a result of consuming the Western diet without stress exposure. In the WD rats, 13 proteins were altered (10 up-regulated and 3 down-regulated). Hemoglobin-related proteins such as Hba1, Hbb, and Hbb-b1 were increased in response to being fed the Western diet. Hemoglobin and its subunits are potent mediators of oxidative responses [40]. One of the mechanisms through which Westernized diets have been shown to elicit their damaging effects is by increasing oxidative stress [41,42]. More specifically, Morrison et al. (2010) suggested that impaired Nrf2 (a key regulator of the antioxidant response) signaling and increased cerebral oxidative stress are the key mechanisms underlying HFD-induced declines in cognitive performance in the aged brain [41]. In the current study, proteins binding calmodulin (Basp1, Akap5; A-kinase anchor protein 5), as well as proteins playing a role in long term synaptic potentiation (Picalm; Phosphatidylinositol-binding clathrin assembly protein, Akap5) or synaptic transmission (Picalm) and formation (Dbnl; Drebrin-like protein), were up-regulated in the temporal cortices of WD rats. The relationship between chronic changes in brain calcium homeostasis and metabolic dysfunctions such as hyperglycemia has been reported in obese mice; the signaling pathways dependent on the activation of Ca^2+^/calmodulin (CaM)-dependent protein kinase II (CaMKII), which was induced by mitochondrial reactive oxygen species, and Ca^2+^ release from ER [43,44,45].

Secondly, the proteomic data clearly show the potent effect of stress exposure and the modification of this effect by being fed with the Western diet. While the predominant stress-induced alterations of protein levels were their down-regulation, the vast majority of proteins significantly changed in the WD/CS rats showed an increased expression as compared to the CS rats. Results from our previous study emphasized that the combination of consuming the Western diet and stress exposure for 6 weeks was more significant with regard to changes in the brain proteome profile than for either of the factors separately [25]. Interestingly, in the present study, both in down-regulated proteins and up-regulated proteins after exposure to stress, the diet combined with stress affected the direction of changes, causing the expression level of most proteins in the WD/CS group to not significantly differ from the control. According to the brain proteome profile, we propose that the observed differences between 6 and 12-week experiments may be related to compensation. Our findings support the hypothesis that the overconsumption of food containing sugar and fat ingredients (comfort food) “compensate” for the effects of chronic stress. Previously, it was reported that a chronic hypocaloric diet reversed some features of stress-induced anxious behaviors [46] or reduced the activity of the HPA axis [47].

Hippocampal proteomics and metabolomics changes induced by stress have been widely described in various animal models (for example, chronic mild stress, chronic unpredictable mild stress, chronic social defeat stress). Impairment in amino acid metabolism and protein synthesis/degradation, dysregulation of glutamate and glycine metabolism, disturbances in fatty acid, abnormal expression of synapse-associated proteins, and intracellular second messenger/signal transduction cascades alterations were revealed [48,49,50]. The hippocampus established the gateway into much of what we have learned about stress and brain plasticity, but in the present study, our goal was to expand the knowledge to the interconnected brain region, which is the temporal cortex.

Chronic stress in experimental studies was linked to imbalances in excitatory and inhibitory neurotransmission in the prefrontal and frontal cortex [51,52]. Here, the authors highlighted proteins that were functionally annotated as involved in axonogenesis, synaptic transmission, and CNS development and differentiation, as well as autophagy and metabolism. The observation of a significant decrease in the cerebrocortical level of Nptx1 suggests stress-induced changes in the modulation of synaptic transmission as well as neurodegeneration processes due to the dual function of this protein. Nptx1 is widely distributed throughout the synaptic terminals of the hippocampus and cortex and is constitutively released from excitatory terminals [53]. Additionally, Nptx1 has been implicated in Alzheimer’s disease, where it is present in dystrophic neurites around plaques in the postmortem brain [54], possibly contributing to apoptosis via effects on mitochondria [55]. Another interesting finding is the up-regulation of Ncam1 and Prrt2 in the CS rats. Ncam1 has been investigated in the context of depression, regarding its multicellular organismal response to stress, learning and memory, and modulation of synaptic transmission [56]. In the study by Togichi et al. (2008), the authors conducted DNA microarray analysis of major depression using postmortem prefrontal cortices. The authors noted the down-regulation of NCAM1, which supported the hypothesis that major depression is associated with impaired cellular proliferation and plasticity [57,58]. Prrt2, enriched in the cerebral cortex, cerebellum, substantia nigra, and hippocampus, was recently characterized as a presynaptic protein that interacts with components of the SNARE complex [59,60]. In the hippocampus of rats subjected to CUMS (chronic unpredictable mild stress), the Prrt2 expression level was down-regulated, according to the iTRAQ results [61]. Preclinical studies have reported altered levels of glutamate, synaptic markers, and dendrite formation in rodent depression models following chronic stress procedures reviewed in [62,63]. Here, changes in Nptx1 and Prrt2, which affect glutamatergic transmission and the significant enrichment of glutamatergic and GABAergic synapses KEGG pathways, support the hypothesis that an alteration in the cortical glutamate levels in the GABAergic pathway may contribute to stress-induced mood disorders and depression [64,65].

Third, there was an overlap between proteins regulated by the Western diet and by chronic stress in GO terms, as the 20 biological pathways enriched in chronic stress-regulated proteomes were enriched to a similar extent with a WD group, including neuron projection morphogenesis, the generation of precursor metabolites and energy, the regulation of postsynaptic membrane neurotransmitter receptor levels, proton transmembrane transport, and vesicle-mediated transport in synapse. The fact that many of the proteins altered by the Western diet and stress are involved in metabolism (generation of precursor metabolites and energy) is unsurprising due to the switch from carbohydrates to fats as the primary energy substrate. Additionally, metabolic changes induced by lifestyle modification, such as the chronic consumption of a high-fat/carbohydrate diet or stress exposure, underlie alterations in brain energy metabolism, implying the adaptation and probable impairment of cellular metabolism and promoting neuroinflammation or an increase in oxidative stress processes, reviewed in [66]. In the current study, KEGG also showed that the pathways related to neurodegenerative diseases were most significant, and the proteins involved in response to oxidative stress and mitochondrial metabolism (cytochrome c oxidase, ATP synthase, and NADH dehydrogenase subunits) were enriched in those pathways. Finally, most protein–protein interactions were observed between proteins associated with carbohydrate metabolism and Parkinson’s disease pathways. This is in line with the well-documented fact that global and regional disturbances in the brain metabolism are causative factors of cognitive impairment and play a pivotal role in the development of neurodegenerative diseases, reviewed in [67].

## 5. Conclusions

In conclusion, this study presents changes in the cerebrocortical proteome in response to the combination of the Western diet and stress exposure, with one-third of the altered proteins associated with neurodegenerative diseases. This study proves the significant interactions between the Western diet and chronic stress on the proteomic changes in the brain. The obesogenic diet and/or stress exposure affects proteins engaged in energy metabolism and fundamental CNS functions. This finding elucidates the association between the appearance of these lifestyle factors and the further risk of cognitive disorders. The female rats used here developed particular obesity phenotypes under simultaneous exposure to the Western diet and stress. This is associated with the distinct proteomic composition revealing the specific brain changes under obesity modulated by stress.

## Figures and Tables

**Figure 1 nutrients-14-01934-f001:**
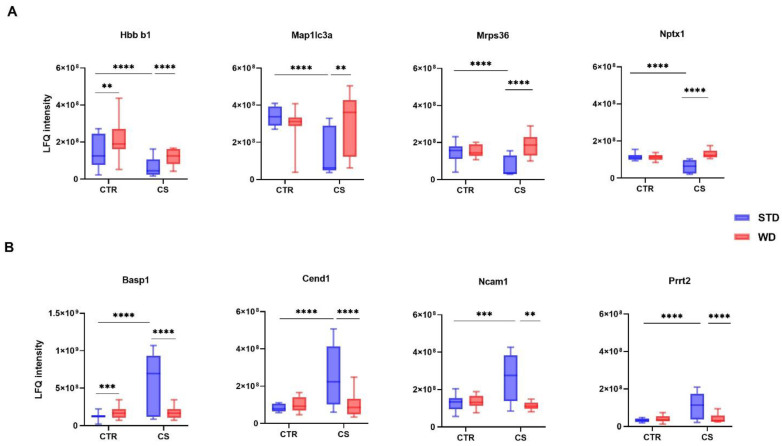
Eight representative proteins showing the most significant changes in the cerebrocortical proteome after exposure to chronic stress and Western diet. The data shown in panels (**A**,**B**) were collected and analyzed from 4 experimental groups: control rats fed with standard diet (STD) (CTR, *n* = 12), rats fed with Western diet (WD, *n* = 12), rats fed with standard diet and subjected to chronic stress (CS, *n* = 6), rats fed with Western diet and subjected to chronic stress (WD/CS, *n* = 6). LFQ = label-free quantitation. Hbb b1 (Hemoglobin subunit beta-1), Map1lc3a (Microtubule-associated proteins 1A/1B light chain 3A), Mrps36 (Mitochondrial ribosomal protein S36), Nptx1 (Neuronal pentraxin-1), Basp1 (Brain acid soluble protein 1), Cend1 (Cell cycle exit and neuronal differentiation protein 1), Ncam1 (Neural cell adhesion molecule 1), Prrt2 (Proline-rich transmembrane protein 2). Data are expressed as median +/− min. to max. Two-way ANOVA, post hoc test: Significant differences are indicated by ** *p* < 0.01, *** *p* < 0.001, **** *p* < 0.0001.

**Figure 2 nutrients-14-01934-f002:**
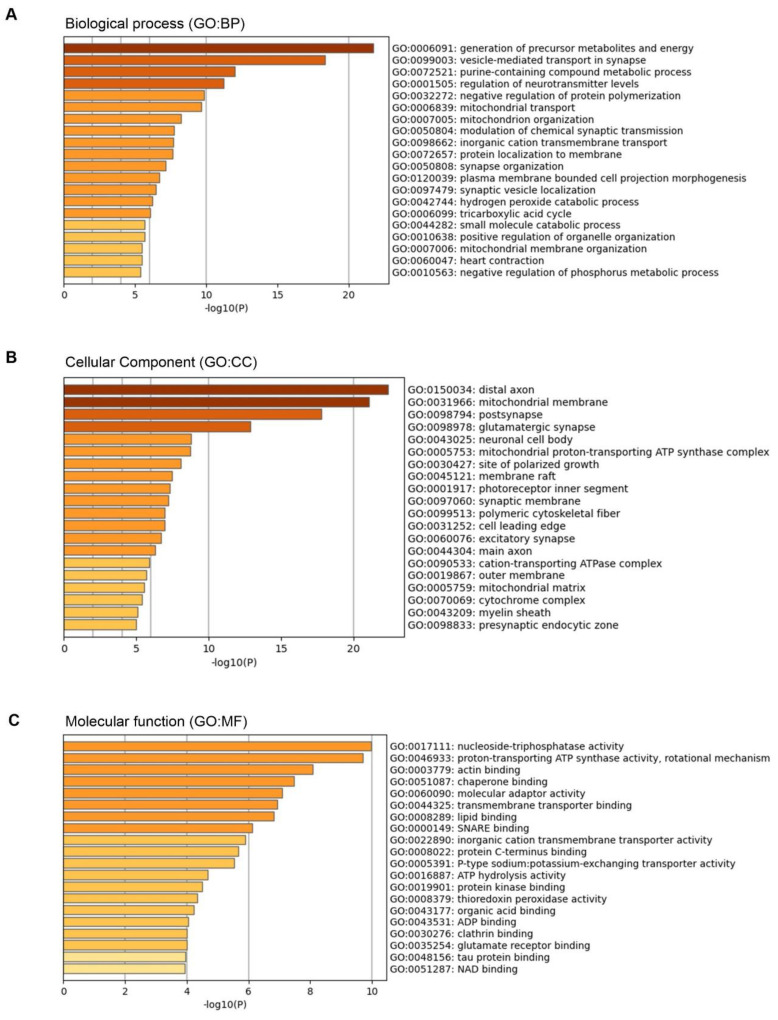
Summary of significantly enriched (*p* < 0.05) biological processes (GO:BP) (**A**), cellular components (GO:CC) (**B**), and molecular functions (GO:MF) (**C**) connected with all identified proteins – results from Metascape (A Gene Annotation and Analysis Resource).

**Figure 3 nutrients-14-01934-f003:**
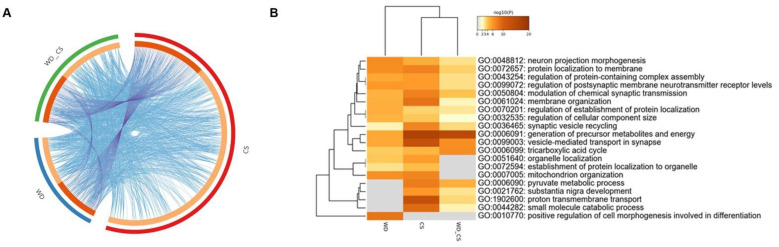
Enriched terms identified for proteins that were significantly different in the WD and CS/WD groups (both in comparison to CTR), filtered and hierarchically clustered (Metascape). (**A**) The circos plot shows how genes from the input gene lists overlap. On the outside, each arc represents the identity of each gene list (blue: WD; red: CS; green: WD/CS). On the inside, each arc represents a gene list, where each gene has a spot on the arc. Dark orange color represents the genes that appear in multiple lists, and light orange color represents genes that are unique to that gene list. Purple lines link the same genes that are shared by multiple gene lists. Blue lines link the different genes where they fall into the same ontology term (Metascape). (**B**) Enriched terms (Biological processes, GO: BP) identified for proteins that were significantly different in the WD, CS, and WD/CS groups (all in comparison to CTR), filtered, and hierarchically clustered (Metascape).

**Figure 4 nutrients-14-01934-f004:**
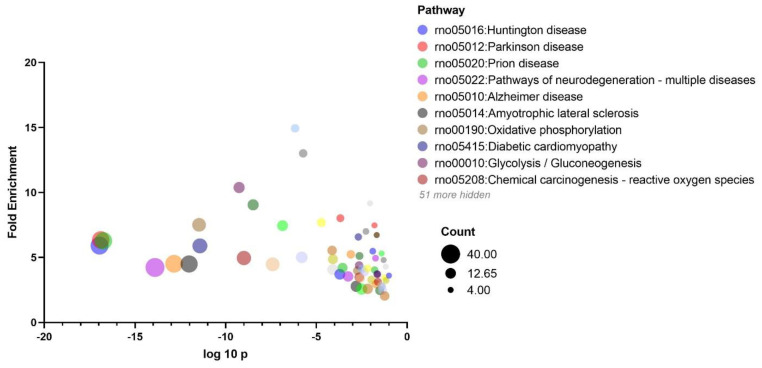
Bubble chart shows KEGG analysis of differential genes. The vertical axis represents the rich factor (ratio of the sum of differential genes enriched in a pathway to the number of genes annotated by the pathway). Bubble size indicates the number of proteins included in each pathway, and different colors indicate different pathways.

**Figure 5 nutrients-14-01934-f005:**
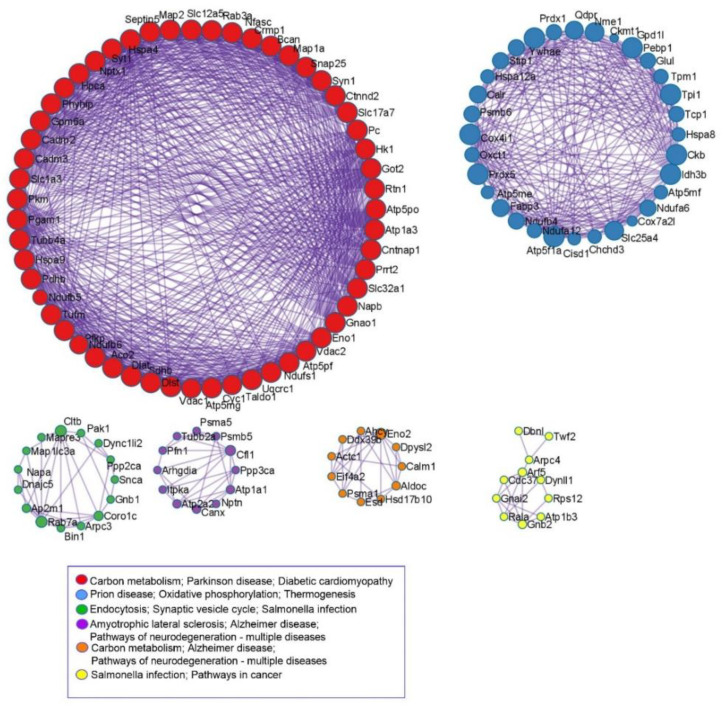
Protein–protein interaction (PPI) networks visualized by Cytoscape software (Institute for Systems Biology, Seattle, WA, USA). All protein–protein interactions among input genes were extracted from PPI data source and formed a PPI network. GO enrichment analysis was applied to the network to extract “biological meanings”. Then MCODE algorithm was applied to this network to identify neighborhoods where proteins are densely connected. The explanation of abbreviations of proteins are listed in Appendix A.

**Table 1 nutrients-14-01934-t001:** Phenotypic parameters of rats subjected to a Western diet and/or chronic stress.

Groups	CTR	WD	CS	WD/CS	Significance		
					Interaction	Diet	Stress
**Energy intake (kcal/day)**	65.95 ± 9.23	103.18 ± 16.64 ****	60.78 ± 21.14	96.04 ± 43.7 ****^,####^	ns	*p* < 0.0001	ns
**Final body weight (g)**	273.99 ± 23.69	359.90 ± 29.89 ****	278.11 ± 10.08	325.75 ± 33.80 **^,#^	*p* = 0.0482	*p* < 0.0001	ns
**Adipose (g)**	10.14 ± 1.80	41.11 ± 12.56 ****	7.03 ± 1.94	21.92 ± 6.98 *^,#,&&&^	*p* = 0.0075	*p* < 0.0001	*p* = 0.0004
**Trunk (g)**	132.12 ± 10.19	151.86 ± 17.02 **	134.62 ± 5.10	150.66 ± 16.26 *	ns	*p* = 0.0007	ns
**Heart (g)**	0.93 ± 0.11	1.09 ± 0.16	0.91 ± 0.04	1.11 ± 0.15	ns	ns	ns
**Liver (g)**	10.71 ± 1.20	11.41 ± 1.86	11.5 ± 2.34	13.13 ± 3.99	ns	ns	ns

Data presented as the mean ± SD, control rats fed with standard diet (CTR, *n* = 12), rats fed with Western diet (WD, *n* = 12), rats fed with standard diet and subjected to chronic stress (CS, *n* = 6), rats fed with Western diet and subjected to chronic stress (WD/CS, *n* = 6), two-way ANOVA, post hoc tests: **** *p* < 0.0001, ** *p* < 0.01, * *p* < 0.05 vs. CTR; ^####^
*p* < 0.0001, ^#^
*p* < 0.05 vs. CS; ^&&&^
*p* < 0.001 vs. WD, ns = not significant.

**Table 2 nutrients-14-01934-t002:** Biochemical and hormonal parameters serum of rats subjected to a Western diet and/or chronic stress.

Groups	CTR	WD	CS	WD/CS	Significance		
Serum					Interaction	Diet	Stress
**Glucose (mg/dl)**	296.8 ± 33.36	317.3 ± 34.97	335.8 ± 46.54	512.98 ± 137.36 ***^,##,&&^	*p* = 0.020	*p* = 0.004	*p* = 0.001
**Insulin (ulU/mL)**	22.57 ± 2.92	23.13 ± 0.38	24.54 ± 5.09	21.68 ± 3.36	ns	ns	ns
**QUICKY index**	0.262 ± 0.002	0.258 ± 0.003	0.256 ± 0.01	0.248 ± 0.008 *	ns	*p* = 0.049	*p* = 0.015
**Leptin (pg/mL)**	1742.5 ± 679.3	12897 ± 3107.6 ****	4222.3 ± 2489.1	1412.6 ± 80.2 ^&&&&^	*p* < 0.0001	*p* < 0.0001	*p* < 0.0001
**Cholesterol (mg/dl)**	79.28 ± 10.89	77.01 ± 10.06	75.03 ± 7.24	75.73 ± 10.38	ns	ns	ns
**LDL cholesterol (mg/dl)**	21.08 ± 4.35	27.71 ± 2.69	17.84 ± 3.03	26.29 ± 4.19 ^#^	ns	*p* = 0.0006	ns
**Triglicerides (mg/dl)**	138.88 ± 21.09	141.88 ± 32.06	192.78 ± 64.6	202.63 ± 69.56	ns	ns	*p* = 0.016
**Estradiol (pg/mL)**	22.17 ± 10.62	20.06 ± 13.20	27.65 ± 15.30	34.14 ± 17.76	ns	ns	ns

Data presented as the mean ±SD, control rats fed with standard diet (CTR, *n* = 8), rats fed with Western diet (WD, *n* = 8), rats fed with standard diet and subjected to chronic stress (CS, *n* = 6), rats fed with Western diet and subjected to chronic stress (WD/CS, *n* = 6), two-way ANOVA, post hoc tests: **** *p* < 0.000, *** *p* < 0.001 vs. CTR; ^##^
*p* < 0.01, ^#^
*p* < 0.05 vs. CS; ^&&&&^
*p* < 0.0001, ^&&^
*p* < 0.01; vs. WD, ns = not siginificant.

## Data Availability

The data that supports the findings of this study are available on request from the corresponding author.

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
