# Peer review of "Global Proteome Profiling of the Temporal Cortex of Female Rats Exposed to Chronic Stress and the Western Diet"

_nutrients, 2022, doi:10.3390/nu14091934_

Round 1
Reviewer 1 Report
The manuscript entitled “Global proteome profiling of the temporal cortex of female rats exposed to chronic stress and the western diet” by Nowacka-Chmielewska and coworkers is focusing an interesting and actual topic related to global proteome profiling of the temporal cortex exposed to chronic stress and the western diet. The overall quality of this work is good and meeting the basic requirement of the journal. The figures look promising and well organized. The technical writing have been prepared well. Literature searching and citation papers are supporting the main content.
I still have some remarks that should be commented by the authors:
- since the hippocampal region is considered as one of the principle in mood regulation connected to cognitive functions, I wonder why it was not considered as a tissue of interest for such an investigation. Please include in the Discussion section.
- also, aside of protein content it seems reasonable to examine the amino acids content alterations in the estimated brain regions. For example, homocysteine levels have been attributed to be involved in numerous brain functions regulation, including anxiety, depression, and cognition. At least it should be commented.
- the behavioral testing concomitantly performed with the obtained analyzes could strongly support the results of the estimated parameters following the applied protocols (rather than comparison to previously published results)
Author Response
Response to Reviewer Katowice, 25th of April 2022
We would like to thank the Reviewer for taking the time and effort necessary to provide all comments and suggestions. We have added all Reviewer’s suggestions to a revision manuscript. The changes in the manuscript are highlighted in yellow. Please find our responses together with the Reviewers suggestions below:
Reviewer #1
The manuscript entitled “Global proteome profiling of the temporal cortex of female rats exposed to chronic stress and the western diet” by Nowacka-Chmielewska and coworkers is focusing an interesting and actual topic related to global proteome profiling of the temporal cortex exposed to chronic stress and the western diet. The overall quality of this work is good and meeting the basic requirement of the journal. The figures look promising and well organized. The technical writing have been prepared well. Literature searching and citation papers are supporting the main content.
I still have some remarks that should be commented by the authors:
- since the hippocampal region is considered as one of the principle in mood regulation connected to cognitive functions, I wonder why it was not considered as a tissue of interest for such an investigation. Please include in the Discussion section.
- The obesogenic diet and weight gain induce many proteomic changes in the central nervous system (Nowacka-Chmielewska et al., 2020, Siino et al., 2018, Smine et al., 2017), but only a few studies have described its impact on hippocampal composition so far. Francis et al., performed the first study where global proteomic changes of the hippocampus were examined after long-term consumption (12 weeks) of an obesogenic diet (Francis et al., 2013). A large number of proteins involved in cellular transport, metabolism, and cell death were altered. On the other hand, hippocampal proteomic and metabolomic changes induced by stress have been widely described in various animal models (chronic mild stress, chronic unpredictable mild stress, chronic social defeat stress) (Yang et al., 2019, Tang et al., 2019, Zhang et al. 2018, Sharanova et al., 2014). Impairment in amino acid metabolism and protein synthesis/degradation, dysregulation of glutamate and glycine metabolism, disturbances in fatty acid, abnormal expression of synapse-associated proteins, intracellular second messenger/signal transduction cascades alterations were revealed. We agree with the Reviewer that the hippocampal region is considered as one of the key regions involved in mood regulation. Also it is unquestionable that the hippocampus provided the gateway into much of what we have learned about stress and brain plasticity but in the present study our goal was to expand the knowledge to the interconnected brain region which is the temporal cortex. According to the Reviewer’s suggestion, we have revised the discussion and we have made every effort to improve the discussion section.
- also, aside of protein content it seems reasonable to examine the amino acids content alterations in the estimated brain regions. For example, homocysteine levels have been
attributed to be involved in numerous brain functions regulation, including anxiety, depression, and cognition. At least it should be commented.
- We agree with the Reviewer's opinion that the measurements of amino acids would be very interesting and shed a light on the received results. We will certainly consider this approach in further research, because to our knowledge there are no such studies. Although potentially very interesting, it is beyond the scope of the present manuscript. According to the Reviewer’s suggestion we have added some commentary to the revised text.
- the behavioral testing concomitantly performed with the obtained analyzes could strongly support the results of the estimated parameters following the applied protocols (rather than comparison to previously published results)
- Thank you. We obviously agree that for further study the behavioral testing should be performed. We are going to continue the studies to examine the effect of stress exposition and western diet on depressive- and/or anxiety-like behaviors.
Reviewer 2 Report
Review Comment
This manuscript by Nowacka et al. explored protein alterations after exposure to the western diet and/or stress in rats model. This is an interesting work that provides information about the impact of the combination of the western diet and stress exposure on cerebrocortical protein alterations and yields insight into the underlying mechanisms and pathways involved in functional and morphological brain alterations as well as behavioral disturbances. Prior to my questions and remarks below, I wish to state that in my opinion the work is suitable for publication following some clarifications and a major revision.
- In Obesogenic rodent diet section, the major ingredients of those sets of snacks such as candy bar (Mars; Mars Inc., McLean, VA, USA), crackers (Lajkonik Snacks, Skawina, Poland), and kabanos) should be given.
- How to estimate whether the chronic social stress model is successful? Is there any biochemical parameters to verify the model is well made?
- In Proteomic analysis procedure, the authors declared LC-MS/MS was performed on a total of rat temporal cortex samples. How many times did the proteomic MS conduct for each group sample, due to repeatability, this experiment should determine in at least triplicate for each sample group.
- Several proteins expression altered in Proteomic analysis. Due to any mild conductions in sample-preparing procedure might affect the cerebrocortical protein alterations, for better solid conclusion, western blot is encouraged to verify the omic results.
- The results declared as “The down-regulation of proteins involved in axonogenesis and mediating the synaptic clustering of AMPA glutamate receptors (Nptx1), as well as proteins related to metabolic processes (Atp5i, Mrps36, Ndufb4), were identified, while increased expression was detected for proteins involved in the development and differentiation of the CNS (Basp1, Cend1), response to stress, learning and memory (Prrt2), and modulation of synaptic transmission (Ncam1, Prrt2).” I wonder whether only few cerebrocortical proteins expression altered as described above. The authors can perform the data within multilayer KEGG or GO analysis, and sum up how many and which proteins altered in a certain function subgroup. It seems impossible that only three proteins Atp5i, Mrps36, Ndufb4 expression changed in metabolic processes. Similarly, only Ncam1, Prrt2 protein expression changed in modulation of synaptic transmission.
Author Response
Response to Reviewer Katowice, 25th of April 2022
We would like to thank the Reviewer for taking the time and effort necessary to provide all comments and suggestions. We have added all Reviewer’s suggestions to a revision manuscript. The changes in the manuscript are highlighted in yellow. Please find our responses together with the Reviewer suggestions below:
Reviewer #2
This manuscript by Nowacka et al. explored protein alterations after exposure to the western diet and/or stress in rats model. This is an interesting work that provides information about the impact of the combination of the western diet and stress exposure on cerebrocortical protein alterations and yields insight into the underlying mechanisms and pathways involved in functional and morphological brain alterations as well as behavioral disturbances. Prior to my questions and remarks below, I wish to state that in my opinion the work is suitable for publication following some clarifications and a major revision.
1. In Obesogenic rodent diet section, the major ingredients of those sets of snacks such as candy bar (Mars; Mars Inc., McLean, VA, USA), crackers (Lajkonik Snacks, Skawina, Poland), and kabanos) should be given.
Ingredients of 2 sets are listed in the description of obesogenic rodent diet in the method section. Additional information regarding macronutrient composition of the snacks used in the present study are stated in the Table S1 A-B.
2. How to estimate whether the chronic social stress model is successful? Is there any biochemical parameters to verify the model is well made?
In the present study, the rats were exposed to the chronic social stress procedure elaborated by Herzog et al. (2009) and modified by our laboratory (Nowacka et al., 2014: doi: 10.1016/j.neures.2014.08.008, Nowacka et al. 2015: doi: 10.1016/j.npep.2015.09.003). Chronic social stress procedure was evaluated for locomotion activity, anxiety-related behaviors, peripheral concentrations of HPA axis hormones (ACTH, corticosterone) and sex steroids (estradiol, testosterone), and the central expression of molecules mediating various aspects of the stress response (Nowacka-Chmielewska et al., 2017: doi: 10.1080/10253890.2017.1376185). The effects of chronic social stress on physiological and behavioral parameters may differ due to rapidly changing ovarian hormone fluctuations. Taking this into consideration, we also evaluated the effects of chronic social stress on the estrous cycle. The chronic stress did not affect the regularity of estrous cycles and circulating concentrations of stress hormones. However, we previously reported a significant increase of ACTH/corticosterone ratio in the stressed female rats, which may reflect lower sensitivity of the adrenal cortex to ACTH. In our previous studies, we have observed the dysregulation of the neuroendocrine response to stress, manifested by region-specific changes in the expression of corticotropin-releasing receptor 1 (CRH-R1) and proopiomelanocortin (POMC).
2. In Proteomic analysis procedure, the authors declared LC-MS/MS was performed on a total of rat temporal cortex samples. How many times did the proteomic MS conduct for each group sample, due to repeatability, this experiment should determine in at least triplicate for each sample group.
This is a misunderstanding, we are sorry for misleading you with this sentence. In fact, each sample was analyzed separately, so for CTR group 11 samples were analyzed, for WD also 11, for CS there were 5 samples, and for CS/WD 6 samples (33 runs in 2 replicates in total). Each sample was processed individually and all further statistical calculations were performed on mean values from group samples peptide/protein intensities normalized in MaxQuant. It allowed for t-test, ANOVA calculations with or without FDR depending on the case.
3. Several proteins expression altered in Proteomic analysis. Due to any mild conductions in sample-preparing procedure might affect the cerebrocortical protein alterations, for better solid conclusion, western blot is encouraged to verify the omic results.
Thank you. We obviously agree that for further study proposed proteins should be confirmed by other techniques, including WB. We are going to continue the studies to perform further validation of potential proteomic biomarkers.
4. The results declared as “The down-regulation of proteins involved in axonogenesis and mediating the synaptic clustering of AMPA glutamate receptors (Nptx1), as well as proteins related to metabolic processes (Atp5i, Mrps36, Ndufb4), were identified, while increased expression was detected for proteins involved in the development and differentiation of the CNS (Basp1, Cend1), response to stress, learning and memory (Prrt2), and modulation of synaptic transmission (Ncam1, Prrt2).” I wonder whether only few cerebrocortical proteins expression altered as described above. The authors can perform the data within multilayer KEGG or GO analysis, and sum up how many and which proteins altered in a certain function subgroup. It seems impossible that only three proteins Atp5i, Mrps36, Ndufb4 expression changed in metabolic processes. Similarly, only Ncam1, Prrt2 protein expression changed in modulation of synaptic transmission.
Here, the authors highlighted proteins which were functionally annotated as involved in axonogenesis (Nptx1), synaptic transmission (Nptx1, Prrt2, Ncam1), CNS development and differentiation (Basp1, Cend1), as well as autophagy (Map1lc3a), and metabolism (Mrps36). Those proteins were chosen among the most significantly down- and up-regulated proteins in response to interventions according to the statistical analysis. Please see supplementary Table S2 which shows the fold change of the identified proteins with the adjusted p-value. In agreement with the Reviewer, in supplementary materials (supplementary data 2) we collected the summarition of KEGG and GO the data showing how many and which proteins altered in a certain function subgroup (please see Appendix B).
Round 2
Reviewer 2 Report
The manuscript has been improved and can be published in the present form.